# Behavior of Male Gamete Fusogen GCS1/HAP2 and the Regulation in *Arabidopsis* Double Fertilization

**DOI:** 10.3390/biom13020208

**Published:** 2023-01-20

**Authors:** Yuka Shiba, Taro Takahashi, Yukino Ohashi, Minako Ueda, Amane Mimuro, Jin Sugimoto, Yuka Noguchi, Tomoko Igawa

**Affiliations:** 1Graduate School of Horticulture, Chiba University, Matsudo 648, Matsudo-shi 271-8510, Japan; 2Graduate School of Life Sciences, Department of Ecological Developmental Adaptability Life Sciences, Tohoku University, 2-1-1 Katahira, Sendai 980-8577, Japan; 3Suntory Rising Stars Encouragement Program in Life Sciences (SunRiSE), Sendai 980-8578, Japan; 4Plant Molecular Science Center, Chiba University, 1-33 Yayoi, Chiba-shi 263-8522, Japan

**Keywords:** GCS1/HAP2, membrane fusion, DMP9, double fertilization, flowering plant

## Abstract

In the sexual reproduction of flowering plants, two independent fertilization events occur almost simultaneously: two identical sperm cells fuse with either the egg cell or the central cell, resulting in embryo and endosperm development to produce a seed. GCS1/HAP2 is a sperm cell membrane protein essential for plasma membrane fusion with both female gametes. Other sperm membrane proteins, DMP8 and DMP9, are more important for egg cell fertilization than that of the central cell, suggesting its regulatory mechanism in GCS1/HAP2-driving gamete membrane fusion. To assess the GCS1/HAP2 regulatory cascade in the double fertilization system of flowering plants, we produced *Arabidopsis* transgenic lines expressing different GCS1/HAP2 variants and evaluated the fertilization in vivo. The fertilization pattern observed in *GCS1*_RNAi transgenic plants implied that sperm cells over the amount of GCS1/HAP2 required for fusion on their surface could facilitate membrane fusion with both female gametes. The cytological analysis of the *dmp8dmp9* sperm cell arrested alone in an embryo sac supported GCS1/HAP2 distribution on the sperm surface. Furthermore, the fertilization failures with both female gametes were caused by GCS1/HAP2 secretion from the egg cell. These results provided a possible scenario of GCS1/HAP2 regulation, showing a potential scheme for capturing additional GCS1/HAP2-interacting proteins.

## 1. Introduction

Double fertilization, a unique reproduction mechanism observed in flowering plants, comprises two independent fusion events between the sperm and egg and the sperm and central cells. Two sperm cells differentiate in each pollen or pollen tube, and the egg and central cells develop in an embryo sac enclosed within the ovule and pistil tissues. Upon pollination, a pollen tube germinates and elongates towards the embryo sac, delivering a pair of sperm cells. After releasing the sperm cells into the boundary of the female gametes, double fertilization occurs through adhesion and membrane fusion between the male and female gametes (Appendix A). Protein regulators on the surface of gametes directly control the processes during double fertilization.

To date, several proteins that function as fertilization regulators in flowering plants have been characterized. The first reported fertilization regulator, GCS1 (GENERATIVE CELL SPECIFIC 1), was identified from a *Lilium longiflorum* generative cell, which is the precursor of the sperm cell [1]. The homologue in *Arabidopsis thaliana* is also called HAP2 (HAPLESS 2), and GCS1/HAP2 was found to be essential for double fertilization [1,2]. The *GCS1*/*HAP2* encodes a single-pass transmembrane protein with a carboxy-terminal transmembrane region, and it is present in the *A*. *thaliana* genome as a single-copy gene. Successive studies have revealed that GCS1/HAP2 function is limited to gamete plasma membrane fusion [3,4,5,6]. Thus, a defect of GCS1/HAP2 causes a complete double fertilization block; *gcs1* sperm cells cannot fertilize both female gametes. GEX2 (GAMETE EXPRESSED 2) was found to be a single-pass transmembrane protein expressed in male gametic cells [7] and has been shown to regulate attachment to the female gamete membrane before membrane fusion [8,9]. According to recent studies, DMP9 (DOMAIN OF UNKNOWN FUNCTION 679 MEMBRANE PROTEIN 9), a four-pass transmembrane protein specifically expressed in sperm cells, is significantly involved in the fertilization of egg cells [10,11]. The nearly identical paralog, DMP8, is known to function redundantly with DMP9 [11]. The biased fertilization failure caused by *dmp8dmp9* sperm cells has provided a clue to elucidate precisely the two independent fertilization events progress in flowering plants.

The molecular mechanisms of double fertilization were unveiled in a step-wise manner during the discovery of each fertilization regulator. As part of the GCS1/HAP2 function scenario, EC1 (EGG CELL 1), a secretory peptide from egg cells, has been reported to activate the fertilization ability of sperm cells [12]. It was suggested that EC1 is involved in GCS1/HAP2 redistribution from the endomembrane systems to the surface of sperm cells for gamete fusion. The GCS1/HAP2 redistribution scenario has been suggested based on an in vitro assay, in which sperm cells in a pollen tube were treated with a mixture of the separately synthesized signature motifs of EC1. A recent study supported this in vitro phenomenon with a similar analysis [13]. Furthermore, the involvement of DMP9 in egg cell fertilization indicates the presence of a regulatory mechanism in GCS1/HAP2-mediated membrane fusion [10,11]. One of the present speculative models is that egg cells are more sensitive than central cells to the abnormal membrane morphology in *dmp9* sperm [14]. Due to DMP1, a paralog protein of DMP9, was reported to be involved in the membrane remodeling of ER and tonoplasts, it has been speculated that DMP9 functions similarly. From these insights and hypotheses, possible DMP9 involvement in the formation of functional GCS1/HAP2 oligomers or in GCS1/HAP2 trafficking from the endomembrane to the plasma membrane by EC1 has been proposed [15]. In our previous study, a delay in egg cell fertilization by DMP9-knockdown sperm cells was observed [10]. The observed delay may suggest that the amount of DMP8 and DMP9 proteins in the plasma membrane could influence the amount of GCS1/HAP2 distributed to the sperm cell surface for fusion, as also speculated by Zhang et al. [15]. Currently, through microscopic imaging, the distribution of GCS1/HAP2 in a pair of *dmp8dmp9* sperm cells in an embryo sac was reported, supporting the involvement of DMP8 and DMP9 in GCS1/HAP2 trafficking to the surface of the sperm cells for membrane fusion; however, why the *dmp8dmp9* sperm cells lacking the surface GCS1 can still fuse with the central cell was not discussed [13]. To understand the precise molecular scenario, further investigations, including reassessment focusing on the collaboration between the identified fertilization regulators, are essential. 

GCS1/HAP2 is known as a bona fide fusogen for gamete fusion between opposite sexes. Amino acid sequence similarities highlighted orthologs not only in plants, but also in protists and invertebrates, with conserved functions in mating or gamete fusion [16,17,18,19,20,21]. The structural homology of GCS1/HAP2 orthologs with class II viral membrane fusion proteins and the somatic fusion family proteins (FF proteins) has been found, indicating that the fusion loops in the GCS1/HAP2 orthologs are essential and sufficient for insertion into the outer layer of the target cell membrane, showing a homo-trimer complex as the post-fusion form [22,23,24,25,26]. The variability in the amino acid residues comprising the fusion loop in the GCS1/HAP2 orthologs suggests species barriers [27], despite the interchangeability of the αF helix region in the fusion loop in flowering plants, *Arabidopsis*, and rice [22]. The viral class II fusion protein is known to drive membrane fusion unilaterally. On the other hand, the bilateral fusion model has been hypothesized in *Tetrahymena*, *Arabidopsis*, and *Plasmodium* mating and fertilization as GCS1/HAP2 expression in both apposed cells is required for efficient membrane fusion [25,28,29]. A recent high-resolution structural analysis of the *Chlamydomonas reinhardtii* GCS1/HAP2 ectodomain revealed a possible region for protein–protein interaction, although the corresponding region is not conserved in land plants [27]. These findings suggest the presence of the GCS1/HAP2-interacting protein for the regulation of fusion activity. 

The fact that only egg cells frequently fail to membrane fusion with sperm cells lacking DMP9 [10,11] also suggests the possibility of the presence of female proteins that regulate GCS1/HAP2 function. The presence of GCS1-interactive proteins in *Arabidopsis* has been implied by the analysis of GAH, a GCS1/HAP2 variant [30]. GAH was constructed by replacing the transmembrane region in GCS1/HAP2 with GFP and, thus, is secreted from the cell [31]. Then, the secreted GAH is expected to interact with the GCS1/HAP2-interacting protein during egg cell maturation. As a result, a double fertilization block was observed when GAH was expressed in the egg cell [30], implying the presence of female proteins that interact with GCS1/HAP2. However, the substantial evaluation of the fertilization block caused by GAH has not been analyzed yet.

Further investigations to fill the gaps between independent insights are needed to understand the GCS1/HAP2 control in membrane fusion during double fertilization. In this study, we performed RNAi, targeting the *GCS1*/*HAP2* to evaluate whether the modified GCS1 expression influenced double fertilization. Second, GCS1/HAP2 distribution in the sperm cell soon before gamete fusion was analyzed, focusing on an arrested *dmp8dmp9* sperm cells in the embryo sac. Moreover, we re-evaluated the possibility of whether the GCS1/HAP2 interactors were present on the surface of female gametes by expressing GAH, a secretion-type of GCS1/HAP2, from the egg cell. The possible molecular scenarios for the gamete membrane fusion by GCS1/HAP2 based on the obtained results are discussed herein.

## 2. Materials and Methods

### 2.1. Plant Material and Transformation

The *Arabidopsis thaliana* HTR10mRFP homozygous line, where the sperm cell nuclei were visualized with mRFP [32], and heterozygous +/*gcs1* mutant with a background of HTR10mRFP (+/*gcs1*^HTR10mRFP^) [8] were used for the transformation experiments in this study. Briefly, seeds were surface-sterilized and germinated on Murashige and Skoog (MS) medium (pH 5.7) supplemented with 1% sucrose (FUJIFILM Wako Pure Chemical Co., Osaka, Japan), 0.5 g/L 2-morpholinoethanesulfonic acid (monohydrate) (NACALAI TESQUE, Inc., Kyoto, Japan), and 0.8% agarose (FUJIFILM Wako Pure Chemical Co., Tokyo, Japan). No antibiotics were used for the germination of HTR10mRFP seeds, while +/*gcs1*^HTR10mRFP^ seeds were germinated with 100 mg/L kanamycin for the selection of *gcs1* loci. Two- to three-week-old seedlings were transferred to the soil and grown at 22 °C under a 16 h light/8 h dark cycle. Then, three to four weeks after acclimatization, a bacterial solution of *Agrobacterium tumefaciens* GV3101 harboring each binary vector was spread on the flower buds for infection [33]. 

### 2.2. Production of the GCS1_RNAi Plant 

The 374–827 bp region in the *GCS1* coding sequence (CDS) (At4g11720) was employed for silencing by RNAi. The 972 bp of the 5’ untranslated region (UTR) was used as the *GCS1* promoter. The promoter, antisense *GCS1*, *GUS* linker, sense *GCS1*, and *NOS* terminator were amplified by adding the appropriate restriction enzyme recognition sites to both ends. The primers are listed in Appendix A. The obtained fragments were cloned into the pUC19 vector and the constructed *GCS1*_RNAi expression cassette between the *Hin*dIII and *Kpn*I sites was transferred into the pPZP221 binary vector [34] (Appendix A). Three separate transgenic T1 plants were used for the evaluation of the seed development phenotype. The ovules from T1 to T3 plants were observed for the evaluation of fertilization patterns. The transgenic line, in which nearly 100% of the T3 seeds, germinated on a medium containing 100 mg/L gentamicin was used for the qRT-PCR analysis.

### 2.3. qRT-PCR Analysis for the GCS1 Expression in the Pollen from GCS1_RNAi 

Total RNA was extracted from the mature pollen of HTR10mRFP plants and each *GCS1*_RNAi line. The flowers at anthesis were collected in 1.5 mL tubes to reach around 1 mL of volume. Afterward, approximately 1 mL of acetone was added, considering the maintenance of pollen viability through dehydration, followed by vortexing for 1 min to liberate the pollen. The acetone solution containing pollen grains was transferred to a new tube, and then centrifuged at 15,000 rpm for 1 min. After removing the supernatant, the pollen was air-dried and kept at −80 °C until RNA isolation. For each transgenic line, total RNA was isolated from five tubes containing the collected pollen using an ISOSPIN Plant RNA kit (Nippon Gene Co., Ltd., Tokyo, Japan). The first-strand cDNA synthesis from 100 ng of total RNA was performed with a kit ReverTra Ace® qPCR RT Master Mix with gDNA Remover (TOYOBO Co. Ltd., Osaka, Japan) using oligo(dT) primer. Real-time PCR reactions were performed using KOD SYBR qPCR Mix (TOYOBO) and an Applied Biosystems StepOnePlus Real-Time PCR System. The target sequence of *GCS1* was amplified with the primers GCS1qRTf and GCS1qRTr (Appendix A). *eIFG4* (At3g60240), as an internal standard, was also amplified with the primers AteIFG4f and AteIFG4r (Appendix A).

### 2.4. Production of the dmp8dmp9^HTR10mRFP^ Plant

The 109–131 bp region in the *DMP9* (At5g39650) ORF was selected as the target sequence for genome editing by CRISPR/Cas9 and cloned into the binary vector, pKI1.1R [35]. The constructed T-DNA region was introduced into HTR10mRFP plants through *Agrobacterium*-mediated transformation. The mutation pattern in the *DMP9* of the transgenic lines was confirmed by sequencing the genomic PCR product amplified with the primers dmp9_GE(HTR10)_F and dmp9_GE(HTR10)_R (Appendix A). A *dmp9*^HTR10mRFP^ line with a homozygous 22 bp deletion (Appendix A) in *DMP9*, but without the T-DNA region in the genome, was selected for T3 generation. The established *dmp9*^HTR10mRFP^ line was crossed with a *dmp8* null mutant line (SALK_13115) provided from Nottingham Arabidopsis Stock Centre (Appendix A). Plants with both homozygous mutant loci were selected from F2 progenies based on amplicon patterns using genomic PCR. Further, the mutation was confirmed by sequencing. The primers used for the above procedures are listed in Appendix A. Finally, the homozygous *dmp8dmp9* plants expressing *HTR10mRFP* in all pollen were selected from the F3 plants, and the re-evaluated seed development phenotype was similar to the previous studies (Appendix A) [11,13].

### 2.5. Production of the GCS1-mNG^dmp8dmp9•HTR10mRFP^ Plants for Observation of GCS1 Distribution in an Embryo Sac

The GCS1-mNG marker was generated by fusing the 1.7-kb *HTR10* promoter, genomic *GCS1*, and *mNeongreen* (mNG; Allele Biotechnology, San Diego, CA, USA) in a pMDC99 binary vector [36]. The mNG was inserted after 685th aa (Val) of GCS1 with a linker sequence (ArgSerArgAsp). Therefore, the last 18 aa was replaced with mNG, as similar to the 14 aa replacements of GFP in the previous GCS1-GFP marker [31]. The TET9-mNG marker was constructed by inserting the 1175 bp *HTR10* promoter, the full-length coding region of *TET9* (At4g30430) CDS, mNG, and *NOS* terminator into a pMDC99 binary vector. The obtained *pHTR10*::*GCS1*-*mNG* and *pHTR10*::*TET9*-*mNG* (Appendix A) were introduced into *dmp8dmp9*^HTR10mRFP^ and +/*gcs1*
^HTR10mRFP^ plants through *Agrobacterium*-mediated transformation, respectively. The GCS1-mNG*^dmp8dmp9^*^•HTR10mRFP^ and TET9-mNG*^dmp8dmp9•^*^HTR10mRFP^ transgenic lines, in which the nuclei RFP and the membrane mNG signals in the sperm cells were observed in all pollen, were selected from the T2 progeny. The GCS1-mNG*^+/gcs1^*^•HTR10mRFP^ and TET9-mNG*^+/gcs1^*^•HTR10mRFP^ transgenic T2 seeds germinated under the pressure of 100 mg/L kanamycin and 30 mg/L hygromycin. The acclimatized plants in which more than 90% of pollen grains contain the sperm nuclei RFP and the membrane mNG signals were used to evaluate seed development (see Section 2.7).

### 2.6. Production of GAH, ssGFP, OsGAH Plants

To construct a binary vector harboring *GAH* for expression in the egg cell, the *CaMV35S* promoter, *EC1.1* promoter [12], *A. thaliana ADH* 5’UTR (At1g77120) [37], and *GAH* [31] regions were amplified using the primers listed in Appendix A. The obtained fragments were fused through the In-Fusion® reaction (TaKaRa Bio, Co. Ltd., Shiga, Japan) and cloned into the *Nco*I and *Pst*I sites in the pPZP221 binary vector, where the *NOS* terminator was inserted in between the *Pst*I and *Hin*dIII sites, resulting in the *pEC1.1*-*ADH* 5’UTR::*GAH* vector (Appendix A). 

To construct *pEC1.1*-*ADH*5’UTR::*ssGFP*, the *GFP* and *NOS* terminator regions were amplified and cloned between the *Eco*RI and *Mau*BI sites in the *pEC1.1*- *ADH*5’UTR::*GAH* vector using the In-Fusion® reaction (Appendix A). 

The CDS of *Oryza sativa GAH* (*OsGAH*), which was optimized for translation in *Arabidopsis*, was synthesized by adding *Pst*I and *Xba*I sites at the 5′ and 3′ ends, respectively (Eurofins Co. Ltd., Tokyo, Japan). The *pEC1.1*-*ADH*5’UTR region was amplified by adding *Sac*I and *Xba*I sites at the 5′ and 3′ ends, respectively. The obtained fragment was inserted between the *Sac*I–*Xba*I sites in the pPZP221 binary vector containing the *NOS* terminator through the In-Fusion® reaction. Furthermore, the synthesized *Xba*I-*OsGAH*-*Pst*I fragment was inserted to obtain *pEC1.1*-*ADH* 5’UTR::*OsGAH* (Appendix A). The primers used are listed in Appendix A. 

As the egg cell plasma membrane marker, a fusion gene of the *GFP* and the aquaporin *PIP2a* (At3g53420) was constructed downstream of the *pEC1.1*-*ADH* 5’UTR. The *pEC1.1*-*ADH* 5’UTR region was cloned into a pENTR 3C vector, harboring *GFP*-*PIP2a* [38]. The *pEC1.1*-*ADH* 5’UTR:: *GFP-PIP2a* was transferred into a pGWB1 destination vector [39] through the LR reaction. 

### 2.7. Evaluation of Seed Development

The silique in which the tip turned yellow (approximately 3–4 weeks after anthesis) was dissected under an SZX9 stereomicroscope (Olympus Inc., Tokyo, Japan). The number of fruited (normal), aborted, and undeveloped seeds in each silique was counted. For HTR10mRFP, +/*gcs1*^HTR10mRFP^, and *GCS1*_RNAi T1 plant lines, seeds from 20–30 siliques were analyzed (Figure 1A). In the evaluation of *GCS1*_RNAi T3 homozygous plant, seeds from 10 siliques were counted per plant, and 6 plants were analyzed for every transgenic lines. Similarly, seeds were counted in +/*gcs1*^HTR10mRFP^ in the same evaluation timing. In the GAH analysis, seeds from 6–17 siliques were counted per plant, and 2–3 plants for every T3 transgenic line were analyzed (Figure 3E). In the complementation test, seeds from 10–17 siliques were counted per plant, and 1–3 T3 plants in which sperm nuclei and membrane signals were detected in more than 90% of pollen were analyzed for each transgenic line (Appendix A). 

### 2.8. Observation of the Embryo Sac, Pollen Grain, and Pollen Tube

To observe the gametes during fertilization in vivo, anthers were removed from the flower bud on the day before anthesis. For the evaluation of fertilization patterns, the ovules at 7–9 h after pollination were analyzed. To observe GCS1/HAP2 distribution in an arrested sperm, the ovules were observed 14–15 h after pollination. For observation, ovules were dissected from the pistil under a stereomicroscope. To observe the sperm cells in the pollen tube, pollen germination was performed as reported in [10]. For observation, the gametic fluorescent signals were captured using a BX51, BX53 (Olympus) or a BZ-X810 (Keyence Co., Ltd., Osaka, Japan) epifluorescence microscope. The fluorescence images were captured with a DP–72 digital camera (Olympus) or a CoolSNAP MYO CCD camera (Photometrics, Tucson, AZ, USA), using cellSense (Olympus) or MetaVue/MetaMorph (ver. 7.8, Molecular Devices, Japan) software. The image processing was performed using Adobe Photoshop ver. 23.5.1 (Adobe, Japan) and Image J2 (ver. 2.9.0). The fertilization state was judged according to the RFP signal morphology of each sperm nucleus [40].

### 2.9. Statistical Analysis 

The data were analyzed using a Wilcoxon rank-sum test (http://www.sthda.com/english/rsthda/unpaired-wilcoxon.php) for seed development phenotype and the chi-square test (Excel software) for fertilization patterns.

## 3. Results

### 3.1. Both Female Gametes Can Fuse with the Sperm Cell from GCS1_RNAi Plants

Assuming that the egg cell fertilization is influenced by the amount of the GCS1/HAP2 in the sperm plasma membrane, it was estimated that a reduction in *GCS1/HAP2* expression causes a similar biased fertilization block as *dmp8dmp9* sperm cells. To evaluate this possibility, the knockdown of *GCS1*/*HAP2* caused by RNA interference was applied here. As a result, reduced normal seed development compared to that of the HTR10mRFP plant was observed in three independent T1 transgenic plants of *GCS1_*RNAi lines (Figure 1A). The significantly higher and lower frequencies of undeveloped seeds were observed in all *GCS1*_RNAi lines relative to the HTR10mRFP and +/*gcs1* plants, respectively (Appendix A). The aborted seed phenotypes were also observed in the *GCS1*_RNAi lines, showing significant differences (*p* < 0.05) relative to the +/*gcs1* plant (Figure 1A,B and Appendix A). Similar aborted seeds were found in the male gametic mutants, *gex2* and *dmp9*, owing to the single fertilization of the central cell [8,10,11]. Therefore, we evaluated the fertilization patterns in *GCS1*_RNAi line 2, which showed the highest frequency of abnormal seed phenotypes (Figure 1A). The ovules were dissected and observed at 7–9 hours after pollination, which is the time that several ovules in a pistil undergo double fertilization with the sperm cells delivered by the first pollen tube, but not by the second pollen tube [3]. The fertilization states of each female gamete can be distinguished through the sperm nuclei signal position in the embryo sac and the morphologies; the success and failure of fertilization are reflected by dispersed and condensed RFP signals, respectively [40] (Figure 1C). In this analysis, ovules showing a dispersed RFP signal were evaluated. Most *GCS1*_RNAi ovules showed a double fertilization pattern (93.5%, *n* = 393); however, a single fertilization pattern (6.5%, *n* = 27) was also observed (Figure 1C,D). Focusing on the single fertilization pattern, the egg cell and the central cell fertilization of the *GCS1*_RNAi line occurred at 2.9% (*n* = 12) and 3.6% (*n* = 15), respectively (Figure 1D). This differs from the *dmp9* knockdown by RNAi [10] and the *dmp8dmp9* knockout lines re-evaluated in this analysis (Figure 1D and Appendix A), where a higher fertilization frequency of the central cell rather than that of the egg cell was observed.

In homozygous T3 transgenic plants, the significant difference in comparison with +/*gcs1* plant in aborted seed frequency was only detected in line 2, while the normal seed frequency relative to HTR10mRFP were significantly different in all lines (*p* < 0.05, Figure 1A and Appendix A). The *GCS1* expression levels in the *GCS1*_RNAi lines were analyzed by qRT-PCR using pollen from T3 homozygous plants because enough pollen for analysis needed to be collected from several plants for each transgenic line. However, as a result, a significant reduction in *GCS1* compared to HTR10mRFP was not detected, showing large standard errors (Appendix A). 

### 3.2. GCS1/HAP2 Expressed in the dmp8dmp9 Sperm Cell in an Embryo Sac Suggested Sperm Surface Localization Soon before Fertilization

As *dmp8dmp9* sperm cells retain the ability to attach to female gamete membranes [10,11], the GCS1/HAP2 distribution immediately before gamete fusion was analyzed for the *dmp8dmp9* sperm cells expressing GCS1/HAP2 fused to a blight fluorescent protein, mNeongreen (mNG).

To confirm the functionality of GCS1-mNG, *GCS1-mNG,* driven by *HTR10* promoter, was expressed in the +/*gcs1* plant. As a result, the infertility of the *gcs1* phenotype was significantly recovered in three independent GCS1-mNG^+/*gcs1*•HTR10mRFP^ transgenic lines (Appendix A). On the other hand, no recovery was observed in the TET9-mNG^+/*gcs1*•HTR10mRFP^ transgenic lines, although one plant line produced a higher number of aborted seeds relative to the +/*gcs1* plant (Appendix A). Compared with the HTR10mRFP plant, the full complementation of infertility was not achieved in GCS1-mNG^+/*gcs1*•HTR10mRFP^. As siliques from plants, in which more than 90% of pollen showed both mNG and RFP signals, were evaluated, none-perfect complementation might attribute to the fact that GCS1-mNG was not expressed in all pollen. Alternatively, it could be due to the different GCS1-mNG expression levels in each transgenic line. Nevertheless, the significant complementation of the *gcs1* phenotype in all GCS1-mNG^+/*gcs1*•HTR10mRFP^ transgenic lines indicates that the GCS1-mNG was functional.

Further, we produced a marker line GCS1-mNG*^dmp8dmp9^*^•HTR10mRFP^ to analyze GCS1/HAP2 distribution in the sperm cells that adhered to the female gamete. As the analytical control, TET9-mNG*^dmp8dmp9^*^•HTR10mRFP^ was also produced as a plasma membrane marker (Appendix A) [41]. In mature pollen, GCS1/HAP2 was preferentially distributed around the sperm cell nuclei, in addition to the cytoplasm (Figure 2A and Appendix A). Dot-like signals reflecting the distribution to the endomembrane system were observed, as reported in previous studies [12,42]. A similar distribution pattern was also observed in the pollen tube (Appendix A), whereas DMP9 showed a distinct outline reflecting the sperm plasma membrane in the pollen and pollen tube (Appendix A) [10]. At 14–15 h after pollination with the GCS1-mNG*^dmp8dmp^^9•^*^HTR10mRFP^ pollen, which is the time that most ovules had accepted at least one pollen tube and completed double fertilization [3], an arrested sperm cell was observed at the boundary of the female gametes (Figure 2B). Among the 21 captured images of the embryo sac containing a single arrested sperm cell with the retained GCS1-mNG signal, a clear mNG signal with a spatial gap from the nucleus RFP signal was observed in 14 embryo sacs (66.7%) while a preferential perinuclear mNG signal was observed in 7 embryo sacs (Figure 2B and Appendix A). In the same analysis with the TET9-mNG*^dmp8dmp9^*^•HTR10mRFP^ pollen, 70% of the sperm cells arrested alone in the embryo sac showed a mNG signal away from the nucleus (*n* = 10) (Appendix A).

### 3.3. Artificially Secreted GCS1/HAP2 from the Egg Cell Hampers Fertilization

In the analyses thus far, *GCS1*_RNAi sperm cells showed no preference between the female gametes in the case of single fertilization (Figure 1D), and GCS1-mNG in *dmp8dmp9* sperm cells was suggested to distribute on the plasma membrane prior to membrane fusion (Figure 2). Considering the previous insight that only egg cells frequently fail to membrane fusion with sperm cells lacking DMP9 [10,11], the presence of a female protein that regulates GCS1/HAP2 function is hypothesized. To further evaluate this possibility, the scheme of GAH (Figure 3A,B), which implied the presence of GCS1/HAP2-interacting protein [30], was re-assessed numerically in this study.

Transgenic lines in which *GAH* was expressed under the control of the *EC1* promoter and a translational enhancer, *AtADH*5’-UTR, were produced with the genetic background of HTR10mRFP (Appendix A). In ovules of the *GAH* lines, condensed RFP signals arrested in an embryo sac were frequently observed, indicating blocked fertilization (Figure 3C). The three independent *GAH* lines showed a significantly higher frequency of undeveloped seeds (19.9–42.8%; Figure 3D,E) than the original HTR10mRFP plant (1.1%; Figure 1A). The aborted seed phenotype implying the single fertilization of the central cell was occasionally observed (0.7–2.4%); at the *p* < 0.05 level, a significant difference to that of in HTR10mRFP (0.4%) was detected in the line 1 (*p* = 4.50 × 10^−4^) and line 3 (*p* = 0.011) and was not in line 2 (*p* = 0.840). As the control, *ssGFP* lines, where a fusion gene of the *GCS1/HAP2* signal sequence and *GFP* was expressed in the egg cell using the *EC1* promoter and *AtADH* 5’-UTR translational enhancer was produced (Appendix A). In *ssGFP* lines, the frequencies of normal, aborted, and undeveloped seeds were 99.1–99.5%, 0.4–0.6%, and 0.2–0.4%, respectively (Figure 3E), indicating no significant difference relative to the original HTR10mRFP plant in all phenotypes (Figure 1A).

Fertilization in the *GAH* (lines 2 and 3) and the *ssGFP* (lines 1 and 2) plants was analyzed at 7–9 h after pollination (Appendix A). The fertilization patterns were judged based on the sperm nucleus morphologies [40]. In this evaluation, one condensed RFP signal pattern was not further categorized into each female gamete fertilization. As a result, approximately half the ovules containing sperm nuclear RFP signals showed blocked fertilization in *GAH* lines. The frequencies of single fertilization and double fertilization blocks were 8.4–11.4% and 38.6–41.0%. These values were significantly higher than those of *ssGFP* lines (*p* < 0.05), even if some cases of two condensed signals reflected the sperm nuclei in a pollen tube unruptured yet. Therefore, the aborted seed observed in *GAH* lines (Figure 3D,E) were probably caused by the single fertilization of the central cell.

To assess the influence of artificial egg cell-expressed GAH on double fertilization, a GAH variant gene constructed based on the *Oryza sativa* GCS1/HAP2 sequence (*OsGAH*) was introduced into the HTR10mRFP plant (Appendix A). For this analysis, the *OsGAH* nucleotide sequence was optimized considering the codon usage in *Arabidopsis* cells and was expressed under the control of *EC1* promoter and *AtADH* 5’-UTR translational enhancer, and *GAH* and *ssGFP* (Appendix A). Three independent *OsGAH* lines showed the normal, undeveloped, and aborted seed phenotypes at frequencies of 97.2–98.1%, 0.5–1.6%, and 1.2–1.3%, respectively (Figure 3D,E). No significant difference in seed development was detected in all *OsGAH* lines compared to HTR10mRFP plants (Figure 1A) and *ssGFP* lines.

## 4. Discussion

The analysis using *GCS1*_RNAi showed the lowered fertilization ability of the sperm cells (Figure 1). On the other hand, a significant reduction in *GCS1* expression was not successfully detected in the *GCS1*_RNAi transgenic lines by qRT-PCR (Appendix A). Nevertheless, significantly abnormal seed development was observed in three independent *GCS1*_RNAi T1 transgenic lines relative to the control HTR10mRFP plant. The obtained qRT-PCR result could be due to the difficulty of analysis for the genes expressed at a low level of gametes enclosed with gametophytic tissue. For example, expression reduction through the knockout of the zygote-specific genes was not detected by qRT-PCR, while an apparent mutant phenotype was reported [43]. In the present analysis, pollen from T3 transgenic plants was used for qRT-PCR analysis to obtain enough pollen samples. Among the T3 plant lines, two lines showed comparable frequencies of aborted seed relative to +/*gcs1*, while the significant differences were detected in all T1 plant lines (*p* < 0.05, Figure 1A and Appendix A). This may reflect the abnormal fertilization caused by the altered *GCS1* level that has hindered the inheritance of such traits to the next generations. Therefore, the differences could have been more undetectable in pollen from the T3 generations. The relation between the GCS1 amount and fusion ability needs to be quantitatively analyzed in the future. 

As single fertilization occurred in the *GCS1*_RNAi lines at higher frequencies than the +/*gcs1* plant (Figure 1A,C), it was implied that the sperm cell with more than the amount of GCS1/HAP2 required for fusion (an amount of GCS1 that is barely enough for membrane fusion) could fuse with each of the female gametes. Moreover, it seemed that the amount of GCS1 does not influence the preference between the female gametes (Figure 1D). Since two sperm cells in each pollen are genetically identical, single fertilization patterns could reflect that the modified GCS1 level irregularly causes it without preference between the distinct female gametes. In addition, both female gametes could have similar sensitivities to the amount of GCS1/HAP2. The next challenge is to develop analytical tools to determine the required amount of GCS1 for membrane fusion.

In our observation, the GCS1-mNG signal was preferentially detected at the perinuclear in addition to the dot-like in the cytoplasm of the sperm cells in pollen (Figure 2A and Appendix A), as reported in a previous study [12]. The GCS1/HAP2 distribution pattern differed from DMP9 and TET9, a sperm plasma membrane localizing protein (Appendix A) [10,11]. In *dmp8dmp9* sperm cells that were arrested in an embryo sac, a clear GCS1-mNG signal away from the nucleus was observed, suggesting the plasma membrane distribution (Figure 2B and Appendix A). Although this study did not evaluate the direct involvement of EC1, the results supported that a GCS1/HAP2 distribution shifted to the plasma membrane in the sperm cells prior to gamete fusion. On the other hand, the observed images (Figure 2B and Appendix A) do not support the previous report that suggested the DMP8 and DMP9 contribution in the GCS1/HAP2 redistribution [13]. The discrepancy might contribute to the differences in observed timing from pollination. In the present analysis, a single sperm cell arrested in an embryo sac was analyzed after 14-15 HAP, which is later than in the previous study (8-9 HAP [13]). 

The results (Figure 2B and Appendix A) also raise a possibility that female gamete plasma membranes, especially those of egg cells, have a DMP9-dependent mechanism to control merging with sperm plasma membranes, even if the enough amount of GCS1/HAP2 for fusion is on the sperm surface. The *Arabidopsis* GCS1/HAP2 was reported to form a trimer at the liposome [22], and DMP8 and DMP9 participation in the GCS1/HAP2 oligomer formation was hypothesized [15]. Scrutinizing the protein dynamics and the amount of GCS1/HAP2 sperm released into an embryo sac after the analysis of the interactions with other fertilization factors would be necessary for the future. However, the establishment of the analytical tools that capture the weak membrane signal in the embedded tissue is a prerequisite.

In the previous study, the egg cell-expressed GAH showed blocked fertilization with wild-type sperm cells, implying the presence of the GCS1/HAP2 partner in female gametes [30]. The present study re-evaluated this phenomenon in more detail (Figure 3). As a result, only the *GAH* lines showed significant fertilization failure, whereas *ssGFP* and *OsGAH* did not. Moreover, single fertilization was observed in *GAH* lines at a significantly higher frequency than that of *ssGFP* (Appendix A). Therefore, competition with the wild-type GCS1/HAP2 was suggested. Furthermore, unhampered fertilization by *Oryza sativa* GAH (Figure 3C,D) may reflect the species specificity of GCS1/HAP2. 

In *Chlamydomonas*, the transition toward GCS1/HAP2 trimerization requires the completion of membrane attachment through an MAR1-FUS1 interaction between the *minus* and *plus* gametes [44,45]. Regarding *Arabidopsis* GCS1/HAP2, trimerization was observed at the liposome, suggesting its post-fusion form [22]. If *Arabidopsis* GCS1/HAP2 requires gamete attachment to transit toward trimer formation, such as *Chlamydomonas*, GAH could have interfered with oligomer formation, causing the failure of fertilization. Although, in the future, it will be necessary to examine this possibility through in vivo analysis of the GCS1/HAP2 trimerization in flowering plants’ double fertilization, as described above, other possibilities could also be hypothesized at the present stage. Based on the results in Figure 2A, after release from a pollen tube, GCS1/HAP2 is suggested to be distributed on the surface of the sperm cells located at the boundary of the female gametes. While the central cell membrane immediately allows membrane fusion via GCS1/HAP2, the egg cell requires the sperm of DMP8 and DMP9 recognition steps to progress the membrane fusion. Therefore, GCS1/HAP2 fusion dynamics are halted on the egg cell membrane. From these insights, the presence of the GCS1/HAP2 interactor on both female gametes, the egg cell and the central cell, is hypothesized in double fertilization. For this, two scenarios are possible considering the results obtained through analyses in this study: (1) the GCS1/HAP2 partner protein is not distributed to the surface of the egg cell until the recognition of DMP8 and DMP9, or (2) the GCS1/HAP2 is masked with a blocker protein until the egg cell recognizes DMP8 and DMP9. Upon the recognition of DMP8 and DMP9, the blocker is removed (or degraded), and GCS1/HAP2 can interact with the partner protein on the surface of the egg cell. The hypothesized model does not contradict the fact that both sperm cells are able to fuse with both female gametes [46], and DMP8 and DMP9 is equally distributed on the sperm cell surface [10,11]. The double fertilization system for sexual reproduction has made flowering plants the most successful among land plants from an evolutional point of view. It is plausible that flowering plants have evolved more additional GCS1/HAP2 regulation mechanisms for gamete membrane fusion, as shown by the involvement of DMP8 and DMP9.

The contribution of other proteins to the regulation of the GCS1/HAP2 fusion activity has been suggested [27], and our cytological analyses further supports the possible presence of interactors in the female gametes. The cell fusion assay with mammalian BHK cells expressing *Arabidopsis* GCS1/HAP2 showed that both cells’ GCS1/HAP2 expression was required for fusion, suggesting bilateral *Arabidopsis* gamete fusion [25]. However, there are few studies on the in vivo dynamics of *Arabidopsis* GCS1/HAP2 because the location where fertilization occurs, the embryo sac, is difficult to analyze. The investigation of the in vivo dynamics of *Arabidopsis* GCS1/HAP2 would be required to examine the above hypothetic models in the future. To better understand GCS1/HAP2 regulation, the identification of interacting proteins, and the evaluation of biochemical interaction between GCS1, DMP8 and DMP9, and unknown proteins are also necessary. Applying the GAH scheme in the female membrane protein isolation procedure [47], we identified GCS1-interacting proteins through proteomics as part of an ongoing study. The participation of the identified proteins in fertilization is under evaluation.

## Figures and Tables

**Figure 1 biomolecules-13-00208-f001:**
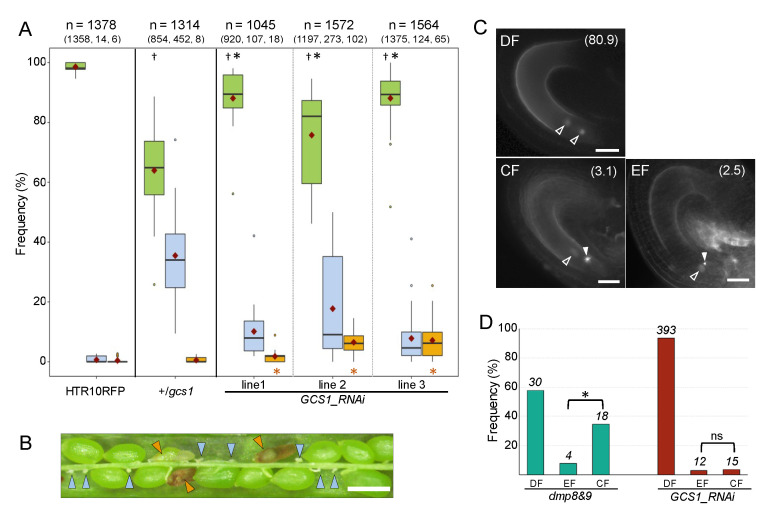
Fertilization failure observed in the *GCS1*_RNAi plants. (**A**) Frequencies of each seed development phenotype observed in HTR10mRFP, +/*gcs1*, and *GCS1*_RNAi T1 transgenic plants. Green, pale blue, and orange boxplots reflect normal, undeveloped, and aborted seed phenotypes, respectively. Seeds from 20–30 siliques were counted per plant. The numbers indicated at the top of each transgenic line are the total counts of the seeds (at 1st line) and each phenotype (at 2nd line in parenthesis). The horizontal bar and violet diamond on each boxplot represent the median and the mean, respectively. The dots associated with each boxplot represent the outliers in the counted data. The dagger and asterisk above the green boxplot represent the significant differences in normal seed phenotype relative to the HTR10mRFP and +/*gcs1* plants, respectively. The orange asterisks below each orange boxplot in *GCS1*_RNAi lines represent the significant differences in the aborted phenotype relative to the +/*gcs1* plant. Significant differences were analyzed with the Wilcoxon rank-sum test (*p* < 0.05). The *p*-values for all comparisons are shown in Appendix A. (**B**) Abnormal seed development observed in the *GCS1*_RNAi plant. Undeveloped and aborted seeds are marked with pale-blue and orange triangles, respectively. Non-labeled green seeds are the normal development phenotype. Bar = 0.5 mm. (**C**) The fertilization patterns observed in *GCS1*_RNAi plant (line 2, T1–T3 plants). Open and filled triangles represent fertilized and arrested sperm cell nuclei signals, respectively. The numbers in parenthesis at the upper-right of the panel are the observed frequencies indicated in the graph in (**D**). DF: double fertilization; EF: egg cell fertilization; CF: central cell fertilization. Bar = 20 µm. (**D**) Frequencies of each fertilization pattern observed in the *dmp8dmp9* and *GCS1*_RNAi plant (line 2). The total number of the counted ovule is indicated in italics above each bar. Asterisks indicate the significant difference detected by the chi-square test (*p* < 0.01). ns: not significant.

**Figure 2 biomolecules-13-00208-f002:**
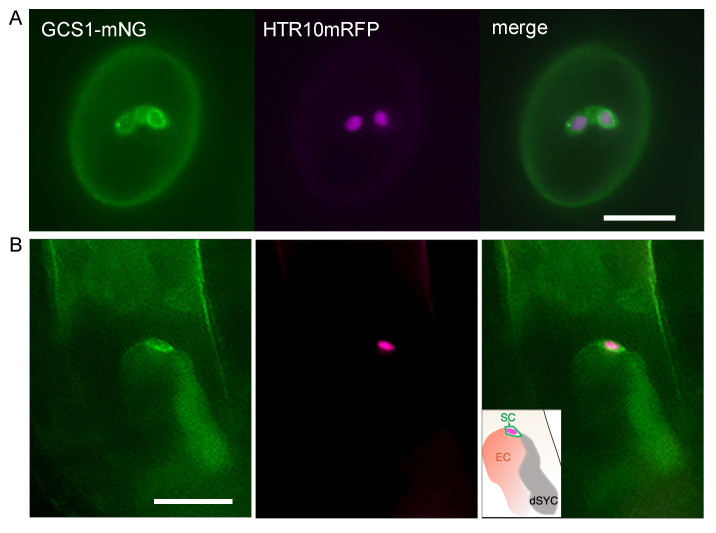
GCS1/HAP2 distribution in the sperm cell before fertilization. (**A**) Fluorescence images of mature pollen from GCS1-mNG*^dmp8dmp9^*^•HTR10mRFP^ plant. GCS1/HAP2 and the sperm cell nuclei are visualized with mNG (left panel) and HTR10mRFP (middle panel), respectively. The right panel is the merged image. Each sperm cell is at a different focus, indicating that GCS1/HAP2 preferentially distributes at the perinuclear (right sperm). The dot-like signals reflect the distribution of the endomembrane system in the cytoplasm (left sperm). The photo of the sperm cells at the same focus plane is indicated in Appendix A. Bar = 10 µm. (**B**) GCS1-mNG in the arrested *dmp8dmp9* sperm cells in an embryo sac. The left, middle, and right panels are the GCS1-mNG, mRFP, and the merged images, respectively. The inset in the right panel represents the schematic explanation of the observed image. SC: arrested sperm cell; EC: egg cell position; dSYC: degenerated synergid cell. Bar = 20 µm.

**Figure 3 biomolecules-13-00208-f003:**
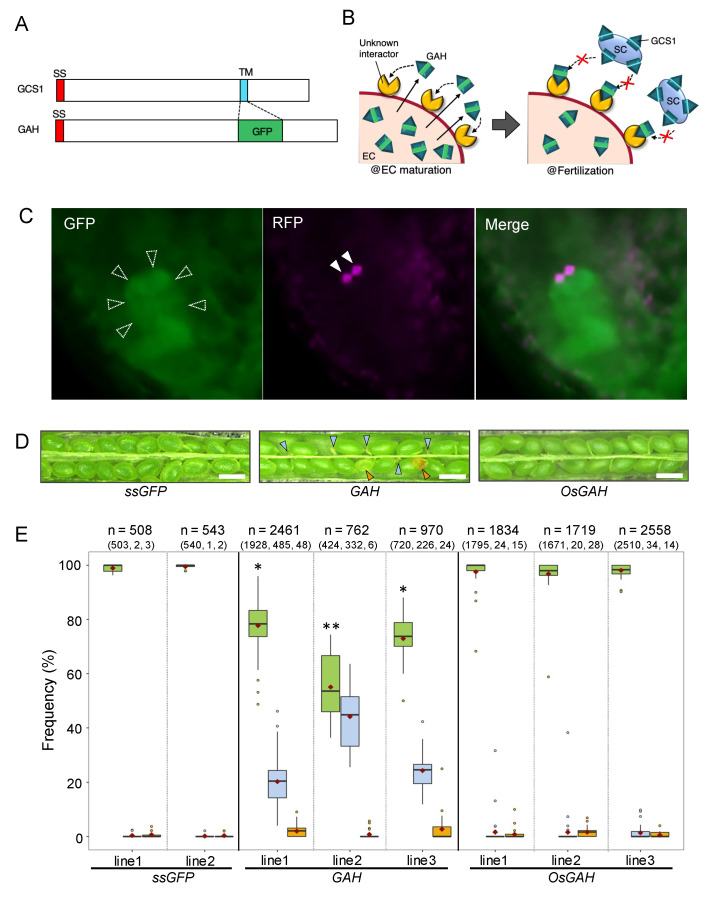
Artificial GAH synthesis in the egg cell blocked fertilization with the normal sperm cells. (**A**,**B**) Pictures are modified from Mori (2014) [30]. (**A**) Schematic view of the GAH construct. The original transmembrane region is replaced with GFP in GAH. SS: signal sequence; TM: transmembrane region. (**B**) The scheme of this analysis. Left: During egg cell maturation, synthesized GAH is secreted from the cell due to the lack of a transmembrane region. The secreted GAH binds to the unknown interactor resident on the cell surface during egg cell maturation. Right: The sperm cells are released from a pollen tube into an embryo sac. GCS1/HAP2 is distributed to the sperm cell surface; however, the male and female gamete membrane fusion does not progress because the GCS1/HAP2 interactor can no longer bind to wild-type GCS1/HAP2. The following fusion process is arrested. SC: sperm cell; EC: egg cell. (**C**) GFP, RFP, and the merged images of an embryo sac from a self-pollinated *GAH* plant. GFP signal derived from the synthesized GAH was detected at the position of the egg cell (dashed arrowheads, left panel). Two magenta signals (triangles, middle panel) are the sperm cell nuclei visualized with HTR10mRFP, indicating that a pair of sperm cells were arrested without fertilization. (**D**) Seed development observed in *ssGFP*, *GAH*, *OsGAH* plants. Undeveloped and aborted seeds are marked with pale blue and orange triangles, respectively. Non-labeled green seeds are the normal development phenotype. Bar = 0.5 mm. (**E**) Frequencies of each seed development phenotype observed in the *ssGFP*, *GAH*, and *OsGAH* lines. The total number of the counted seeds in each plant line is indicated at the top of the column. The horizontal bar and a violet diamond on each box represent the median and the mean, respectively. The dots associated with each boxplot represent the outliers in the counted data. The different number of asterisks indicates the significant differences relative to the HTR10mRFP plant (Figure 1A) according to the Wilcoxon rank-sum test (*p* < 0.01).

## Data Availability

Not applicable.

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
