# Peer review of "Behavior of Male Gamete Fusogen GCS1/HAP2 and the Regulation in Arabidopsis Double Fertilization"

_biomolecules, 2023, doi:10.3390/biom13020208_

Round 1

Reviewer 1 Report

The manuscript of Shiba et al reports the functional analysis of the gene GCS1/HAP2 in relation of gamete fusion in Arabidopsis. Overall, the manuscript could be of interest among the scientific community whose research interests are focused on plant fertilization dynamics. Indeed, experiments are designed quite well and questions raised in the introduction seem to appropriately addressed with the provided evidences. However, in my opinion the manuscript needs substantial revision from both a formal and content points of views.

My major point of criticism deals with the discussion of the outcome of GCS1_RNAi transformation. Based on the RT-qPCR analysis of GC1 in pollen of RNAi interfered plants, no clear evidences of knocked down regulation of mRNAs are provided and nevertheless the authors stated that the fertilization failure in these transgenic lines might be attributed to a reduced GC1/HAP2 protein level. The Authors tried to justify this discrepancy with the high heterogeneity of pollen expression of this gene in pollen as witnessed by the high SD of the samples. Since this as central point of the whole manuscript, I would clarify this discrepancy showing additional evidences (i.e. showing which is the main source of variability of RT_qPCR  experiments, among or within replicates and/or showing additional RT-qPCR outcomes in other tissues). Conclusion on GCS1/HAP2 role on gamete fusion based on these evidences are not acceptable as these stand.  

Secondly, the text and English form are very confusing thus rendering difficult to the reader to follow the logic sequence of text. Just an example, citations are frequently reported even in the results section rendering difficult to distinguish what the Authors did in this research and what derives from previous works. I suggest to revise the manuscript in depth with the aid of mother tongue editing  

Minor remarks

Abstract

Line 14: …two independent fertilization events occur…  

Line 17: Other sperm membrane..

Introduction

Line 43: Please give short definition for “single-pass transmembrane protein” or give ref.

Line 53: …fertilization of egg cell..

Lines 62-63: …the sentence ”..the redistribution …EC1.” makes no sense as it stands, please re-phrase.

Lines 84-85: “In this study….fertilization.” see my major point of criticism above.

Materials and methods

Line 105: 972 bp upstream(?) 5’UTR region

Results

Lines 213-215 and 229-230: see my major point of criticism above.

Discussion

Line 435: never start a new sentence with “And”

Line 450: made flowering plants the most successful among land plants from the evolution point of view.

Line 457: “Although in the future it will necessary to examine…

Author Response

Dear Reviewer 1,

Thank you very much for taking the time to read the manuscript carefully. We revised the manuscript following the reviewer’s comments as much as possible; however, we could not satisfy the reviewer’s requests, especially on qRT-PCR analysis. Regarding this, we describe the reason. We would be grateful if the reviewers could accept the revised version as a ‘Communication’ article.

Point 1: My major point of criticism deals with the discussion of the outcome of GCS1_RNAi transformation. Based on the RT-qPCR analysis of GC1 in pollen of RNAi interfered plants, no clear evidences of knocked down regulation of mRNAs are provided and nevertheless the authors stated that the fertilization failure in these transgenic lines might be attributed to a reduced GC1/HAP2 protein level. The Authors tried to justify this discrepancy with the high heterogeneity of pollen expression of this gene in pollen as witnessed by the high SD of the samples. Since this as central point of the whole manuscript, I would clarify this discrepancy showing additional evidences (i.e. showing which is the main source of variability of RT_qPCR experiments, among or within replicates and/or showing additional RT-qPCR outcomes in other tissues). Conclusion on GCS1/HAP2 role on gamete fusion based on these evidences are not acceptable as these stand. 

Response 1: Another reviewer pointed out a similar concern, and we realize we should carefully describe the present result with appropriate discussion. The answer described below was similarly responded to Reviewer 2.

We could not indicate a significant reduction of GCS1 expression at the molecular level. For the qRT-PCR analysis, we have tried several times with different primer sets for GCS1 and different kinds of standard genes. In the previous manuscript, we showed the result when a sperm cell-specific gene HTR12 was used as the standard. In the revised version, we indicated a different result obtained with primers for eIFG, a constitutive gene, as standard, which was a similar method in our previous qRT-PCR for DMP9 expression (Takahashi et al. 2018; The almost complete reduction of DMP9 was achieved using an entire region of DMP9 for RNAi). Nevertheless, significant reductions were not detected. According to the pollen transcriptome study (Borg et al. 2011, doi: 10.1105/tpc.110.081059), the GCS1 (At4g11720) expression level was approximately five times lower than that of DMP9 (At5g39650) in mature pollen; therefore, it could be one of the reasons that GCS1 reduction in the sperm cells within pollen was not significantly detected. The qRT-PCR is sometimes difficult for the genes expressed specifically in the gamete or zygote enclosed with the gametophytic tissues. For example, the reduction of BBM and PLT2 (embryogenesis regulators) expressions in the knockout mutants were not detected even if the altered phenotypes were observed (Chen et al. 2022; doi.org/10.1073/pnas.2201761119). In the present study, pollen from T3 generations selected based on antibiotic resistance was used for qRT-PCR. However, the reduced GCS1 expression trait that caused fertilization failure might have been difficult to inherit. So, we replaced the counted data indicated in Fig. 1A (seed development) with that obtained from the T1 generations. The counted data of +/gcs1 was also replaced with that obtained by the same person who performed the T1 evaluation at the same timing. Statistical analyses were performed with the T1 data, and the significant differences in aborted seed phenotype were detected relative to +/gcs1 at p < 0.05 level. We added the detected p-values in Table S1. We understand that it was preferred to perform qRT-PCR with T1 transgenic lines, however, it is almost impossible to obtain enough amount of pollen samples from a single transgenic plant.

Western blotting is a more confident evaluation, as reviewer 2 suggested. However, it also could be challenging to detect the mild reduction of GCS1 because enough amount of sperm cell collection is more technically difficult than that of pollen. In addition, according to Dr. Mori, who identified GCS1, the anti-GCS1 antibody for Arabidopsis has not been kept anymore. Because we are given 10 days to reply to the comments, we don’t have time to obtain further transgenic lines. Instead, we revised the descriptions carefully, minding not to be conclusive. We understand that more solid evidence would be needed to clarify the relationship between the GCS1 amount and membrane fusion, also through in vitro analysis. We would be appreciated it if you could agree that the revised descriptions are acceptable as ‘Communication’.

We added information about pollen collection and total RNA isolation in the revised materials & methods. Accompanied by the changes in Fig. 1, we revised the discussion.

Point 2: Secondly, the text and English form are very confusing thus rendering difficult to the reader to follow the logic sequence of text. Just an example, citations are frequently reported even in the results section rendering difficult to distinguish what the Authors did in this research and what derives from previous works. I suggest to revise the manuscript in depth with the aid of mother tongue editing

Response 2: We are sorry for the lengthy and hard-to-read descriptions. The manuscript was revised, minding the points given from reviews. The revised version is submitted after the English editing service.

Minor remarks

Abstract

Line 14: …two independent fertilization events occur… 

Line 17: Other sperm membrane..

Thank you for your kind corrections. Words were revised.

Introduction

Line 43: Please give short definition for “single-pass transmembrane protein” or give ref.

The words “with a carboxy-terminal transmembrane region” as a short description were added.

Line 53: …fertilization of egg cell..

The text was revised.

Lines 62-63: …the sentence ”..the redistribution …EC1.” makes no sense as it stands, please re-phrase.

Accompanied by this revision, the descriptions in the previous section 3.2 were moved here to explain more detail.

Lines 84-85: “In this study….fertilization.” see my major point of criticism above.

The descriptions were revised, minding the construction and the contents.

Materials and methods

Line 105: 972 bp upstream(?) 5’UTR region

We are sorry for the confusing expression. The text was revised, indicating the abbreviation of UTR.

Results

Lines 213-215 and 229-230: see my major point of criticism above.

Descriptions were revised, minding not to be conclusive.

Discussion

Line 435: never start a new sentence with “And”

Text was revised.

Line 450: made flowering plants the most successful among land plants from the evolution point of view.

Line 457: “Although in the future it will necessary to examine…

Thank you very much for the suggestions. Texts were revised.

Reviewer 2 Report

        The manuscript entitled “Behavior of male gamete fusogen GCS1/HAP2 and the regulation in Arabidopsis double fertilization” by Shiba et. al. employed GCS1_RNAi lines, modified GCS1-mNG construct, and competitive GAH (GCS1 without TM expressed in the egg cell) to examine the effect of reduced GSC1 on double fertilization, to monitor GCS1 distribution before double fertilization, and to confirm the existence of a putative interactor of GCS1 on the surface of the egg cell.  I think the ideas are great but the data for the first two are not very solid while for the third one, it’s very interesting but most has been published in 2014 and only silique data and OsGAH is originally contributed here. Please find details below.

Major concerns

1.       For GCS1_RNAi lines/experiments, firstly, the authors cannot claim that less GCS1 level affects double fertilization by data from these RNAi lines since RNA level of each line is not different from WT control (even looks higher). If they believe protein level is reduced, please prove it. Secondly, since the standard errors for qPCR are very big, there seems to be technical issues. Also, please state how you collect pollen for RNA experiment (it may affect your qPCR results). Thirdly, the phenotype is very subtle. If GCS1 RNA/protein level is substantially not different from WT, the phenotype could be due to other reasons, eg, non-specific knockdown. Double check if the sequence used for RNAi is GCS1 specific. Fourth, the aborted phenotype in RNAi lines seem to be not very specific or is environmental dependent, as 1) the aborted phenotype in +/gcs1 is found in Fig 1A, but not in Fig S4; 2) complementation of TET9 in +/gcs1 also show aborted seeds.

To test the effect of less GCS1 on double fertilization, I encourage the authors to screen more RNAi lines (if the sequence used is GCS1 specific) with gradient GCS1 RNA level and then check the phenotype.

2.        The complementation of +/gcs1 by GCS1-mNG seems to be partial rather than full. The percentage of unfertilized+aborted phenotype is almost identical to GCS1_RNAi lines. Please do statistical analysis for the comparison (normal vs abnormal). If the complementation is partial, the dot-like localization may not be true. On the other hand, if the partial complementation of line 2 is due to lower expression of GCS1 (please check the RNA level), it may serve as a knockdown line in answering the first question (effect of less GCS1 on double fertilization) if the -18aa GCS1-mNG is proved to be fully functional.

3.       More high quality images are needed for Fig 2 (GCS1/HAP2 distribution in the sperm cell before fertilization).  “Each sperm cell is at a different focus”, please provide individual images for each sperm cell on focus so that we can better see the localization. For Fig 2B, please provide unmerged images of GCS1-mNG and HTR10mRFP so that we can see the real localization of GCS1. Also need control from WT (non-dmp8/9 mutants) background, so that we can compare the effect of dmp8/9 on the distribution of GCS1-mNG before fertilization.

4.       For Fig 3, the idea of GAH construct, protein distribution and sperm arrest has been published in 2014 (ref 27, https://link.springer.com/chapter/10.1007/978-4-431-54589-7_26).

Minor concerns

1.       Please better explain the logic of several sentences, as it’s hard to understand how the front information can get the latter concern/conclusion/deduction.

2.       Some background information in the Results needs to go to the Introduction section.

3.       Introduce how many lines of each transformed constructs were selected and used in the Method section. This is more important than how many T2/T3 plants were used.

4.       Only one dmp8 and dmp9 mutant line was used in the study, which cannot exclude the concern of T-DNA insertion position or off target effect.

5.       Draw schematic diagrams for each constructs as a supplemental figure for easier understanding.

6.       Cite reference for PIP2a and agro transformation.

7.       Line 232-233 “Such result might reflect the heterogeneous GCS1/HAP2 expression per pollen within the pollen population”, aren’t they homozygous? Check if there are multiple T-DNAs in a line.  Make sure pollen collection method is good.

8.       For Fig 1D, the sample number for EF and CF is too low to conclude bias fertilization or not. Also, the number of unfertilized ovules is much higher in dmp8/9 double mutants than in RNAi lines, suggesting its important and equal function in double fertilization.

9.       Line 241 and others, what do you mean by “over a fusion-required amount”?

10.   Line 282 and more, what do you mean by “linear mNG signal”? Did you mean plasma membrane-like localization?  For this, it’s better to do co-localization with a PM marker. For Fig S6C, the signal of TET9-mNG also looks like “perinuclear”. And, what’s the image in the right bottom corner of Fig S6 for?

11.   Line 350 and more, I think “competition” is more accurate than “antagonistic effect”: GAH (GCS1 without TM from egg cell) competes the unknown interactor with GCS1 with TM in the sperm cell before they arrive. It looks like signaling is triggered in the sperm cells (via GCS1 with TM) once GCS1-DMP8/9-UNKNOWN INTERACTOR interact with each other for egg cell fertilization.

12.   For Fig3C, the authors show two pairs of arrested sperm nuclei, which is usually a less common phenotype compare to single pollen tube (one pair of sperm nuclei). Authors need to clarify if this is the dominant/representative phenotype or not, otherwise it could be misleading to readers.

13.   Line 369 and 376, Figure S4 should be Figure S5. Line 386-387, check the Figure order or the statement, as what provided doesn’t match the statement in front.

Author Response

Dear Reviewer 2,

Thank you very much for the valuable comments to improve our manuscript. We revised it following the reviewer’s comments as much as possible; however, we could not satisfy all of the reviewer’s requests, especially on qRT-PCR analysis. Regarding this, we describe the reason. We would be grateful if the reviewers could accept the revised version as a ‘Communication’ article.

Point 1: My major point of criticism deals with the discussion of the outcome of GCS1_RNAi transformation. For GCS1_RNAi lines/experiments, firstly, the authors cannot claim that less GCS1 level affects double fertilization by data from these RNAi lines since RNA level of each line is not different from WT control (even looks higher). If they believe protein level is reduced, please prove it. Secondly, since the standard errors for qPCR are very big, there seems to be technical issues. Also, please state how you collect pollen for RNA experiment (it may affect your qPCR results). Thirdly, the phenotype is very subtle. If GCS1 RNA/protein level is substantially not different from WT, the phenotype could be due to other reasons, eg, non-specific knockdown. Double check if the sequence used for RNAi is GCS1 specific. Fourth, the aborted phenotype in RNAi lines seem to be not very specific or is environmental dependent, as 1) the aborted phenotype in +/gcs1 is found in Fig 1A, but not in Fig S4; 2) complementation of TET9 in +/gcs1 also show aborted seeds.

To test the effect of less GCS1 on double fertilization, I encourage the authors to screen more RNAi lines (if the sequence used is GCS1 specific) with gradient GCS1 RNA level and then check the phenotype.

Response 1: As reviewer 1 pointed out, we could not indicate the significant reduction of GCS1 expression at the molecular level. For the qRT-PCR analysis, we have tried several times with different primer sets for GCS1 and different kinds of standard genes. In the previous manuscript, we showed the result when a sperm cell-specific gene HTR12 was used as the standard. In the revised version, we indicated different result obtained with eIFG primers as standard, which was a similar method in our previous qRT-PCR for DMP9 expression (Takahashi et al. 2018; The almost complete reduction of DMP9 was achieved using an entire region of DMP9 for RNAi). Nevertheless, significant reductions were not detected. According to the pollen transcriptome study (Borg et al. 2011, doi: 10.1105/tpc.110.081059), the GCS1 (At4g11720) expression level was approximately five times lower than that of DMP9 (At5g39650) in mature pollen; therefore, it could be one of the reasons that mild reduction of GCS1 in the sperm cells within pollen was not significantly detected. The qRT-PCR is sometimes difficult for the genes expressed specifically in the gamete or zygote enclosed in the gametophytic tissues. For example, the reduction of BBM and PLT2 expressions in the knockout mutants were not detected even if the altered phenotypes were observed (Chen et al. 2022; doi.org/10.1073/pnas.2201761119). In the present study, pollen from T3 generations selected based on antibiotic resistance was used for qRT-PCR. However, the reduced GCS1 expression trait that caused fertilization failure might have been difficult to inherit. So, we replaced the counted data indicated in Fig. 1A (seed development) with that obtained from the T1 generations. The counted data of +/gcs1 was also replaced with that obtained by the same person who performed T1 evaluation at the same timing. Statistical analyses were performed with the T1 data, and the significant differences in aborted seed phenotype were detected relative to +/gcs1 (p < 0.05 level). We added the detected p-values as Table S1 (Regarding TET9-mNG complementation test, please see Response 2). We understand that it was preferred to perform qRT-PCR with T1 transgenic lines, however, it is almost impossible to obtain enough amount of pollen samples from a single plant.

We agree that western blotting is a more confident evaluation. However, it also could be challenging to detect the mild reduction of GCS1 because enough amount of sperm cell collection is more technically difficult than that of pollen. In addition, according to Dr. Mori, who identified GCS1, the anti-GCS1 antibody for Arabidopsis has not been kept anymore.    

 Because we are given 10 days to reply to the comments, we don’t have time to obtain further transgenic lines. Instead, we revised the descriptions carefully, minding not to be conclusive. We understand that more solid evidence would be needed to clarify the relationship between the GCS1 amount and membrane fusion, also through in vitro analysis. We would be appreciated it if you could agree that the revised descriptions are acceptable as ‘Communication’.

We added information about pollen collection and total RNA isolation in the revised materials & methods. It was confirmed that none-specific regions of more than 20-mer are not detected within the target GCS1 region employed in this study (by BLAST to Arabidopsis transcripts in the TAIR database).

Point 2: The complementation of +/gcs1 by GCS1-mNG seems to be partial rather than full. The percentage of unfertilized+aborted phenotype is almost identical to GCS1_RNAi lines. Please do statistical analysis for the comparison (normal vs abnormal). If the complementation is partial, the dot-like localization may not be true. On the other hand, if the partial complementation of line 2 is due to lower expression of GCS1 (please check the RNA level), it may serve as a knockdown line in answering the first question (effect of less GCS1 on double fertilization) if the -18aa GCS1-mNG is proved to be fully functional.

Response 2: We added p-value data as Figure S5B in the revised version. As concerned by the reviewer, one TET9-mNG line (no. 3) showed a significant difference (p < 0.01) compared to the HTR10mRFP and +/gcs1. In contrast, all TET9-mNG lines did not show differences in the undeveloped seed phenotype relative to +/gcs1. TET9-mNG was used as the negative control, and it was also the fact that TET9-mNG did not complement the infertility caused by gcs1. The discussion was revised, mentioning the detected exception.

Regarding normal seed phenotype, GCS1-mNG lines did not achieve full complementation compared to the HTR10mRFP, whereas all three lines showed significant infertility recovery relative to +/gcs1. The transgenic lines were selected with kanamycin for gcs1 (T-DNA insertion) and hygromycin for the different T-DNA harboring GCS1 (or TET9)-mNG. Although counted data was recorded with only GCS1-mNG+/gcs1 lines 1 and 2, 98.2 (n = 55) and 90.9% (n = 44) of pollen had mNG signals with nuclei RFP in lines 1 and 2, respectively. We had written “plants in which almost all of pollen grains contain the sperm nuclei RFP and the membrane mNG signals were used” in section 2.5 in the previous manuscript, and here it was revised to “more than 90%”. For plant selection in other lines, “more than 90%” was judged based on a similar image of the pollen population when observed. The none-perfect complementation could have been that mNG was not expressed in all pollen.

Relating to the functionality of GCS1-mNG, it was reported that a mutant GCS1/HAP2, in which GFP was inserted into the C terminal side from the transmembrane region (GPP), was functional (Mori and Hirai et al. 2010; doi:10.1371/journal.pone.0015957). The GCS1-mNG used in our study has the mNG insertion at almost the same position as GPP. The protein structures are similar, although the sizes of GFP (27.23 kD) and mNeongreen (30.73 kD) are slightly different. The previous insights and the fact that significant infertility recovery was observed in three independent transgenic lines make us consider that GCS1-mNeongreen is also functional. Although qRT-PCR analysis is challenging, as responded above, we think obtaining both fluorescence expression and infertility recovery can support the functionality of GCS1-mNG.

Regarding distribution patterns, similar GCS1 dot-like signals in the sperm cells within a pollen or pollen tube have been reported in different studies (Sprunck et al. 2012; doi:10.1126/science.1223944 /Wang et al. 2022; doi.org/10.1073/pnas.2207608119). From this point as well, it is reasonable to consider that GCS1-mNG in this study was functional. The previous descriptions were misleading, so these were revised. We would appreciate the reviewer’s understanding.

Point 3: More high quality images are needed for Fig 2 (GCS1/HAP2 distribution in the sperm cell before fertilization).  “Each sperm cell is at a different focus”, please provide individual images for each sperm cell on focus so that we can better see the localization. For Fig 2B, please provide unmerged images of GCS1-mNG and HTR10mRFP so that we can see the real localization of GCS1. Also need control from WT (non-dmp8/9 mutants) background, so that we can compare the effect of dmp8/9 on the distribution of GCS1-mNG before fertilization.

Response 3: Capturing the weak membrane signal from a tiny sperm cell in an embryo sac embedded with ovule tissue as each Z-section slice with CLSM is very hard; therefore, observations in this study were dared to perform with epifluorescence microscopes (we don't mean CLMS capturing is impossible, but it frequently requires long-exposure and high offset settings, etc., resulting in saturated images). The purpose of indicating Figure 2A was to show the GCS1-mNG detected in both perinuclear and dot-like signals in the sperm cell as reported in the different studies (introduced in response 2). We added photos of the same pollen with the sperm cells at the same focus plane in the revised Figure S6. In the revised Figure 2B, separated mNG and RFP images were added. To understand the correct distribution, immuno-labeling and electron microscopy are critical evaluations in the future, although it's still technically challenging. In that sense, visualization by fusion FPs has an artificial aspect. In the revised version, we tried not to be conclusive, describing that the present data are suggestive.

Since gamete fusion starts within several minutes after being released from a pollen tube, it is impossible to judge which sexual steps two WT sperm cells are in an embryo sac (the previous report by Wang et al. did not exclude this possibility). Therefore, only the evaluation focusing on a GCS1-mNGdmp8&9 sperm that remained alone in an embryo sac was performed in our study because such sperm cells reflect that it completed attachment to the female gamete.

Pint 4: For Fig 3, the idea of GAH construct, protein distribution and sperm arrest has been published in 2014 (ref 27, https://link.springer.com/chapter/10.1007/978-4-431-54589-7_26).

Response 4: In the previous report by Mori, no numerical analysis was performed, although the failure of fertilization was observed. In the present study, we used a different egg cell-specific promoter and a translational enhancer to evaluate the effect numerically. The scheme was already reported, as the reviewer pointed out. Therefore, we added the source information in Fig. 3B caption.

Minor concerns

  1. Please better explain the logic of several sentences, as it’s hard to understand how the front information can get the latter concern/conclusion/deduction.
  2. Some background information in the Results needs to go to the Introduction section.

We are sorry for the lengthy and hard-to-read descriptions. We revised regarding these points.

  1. Introduce how many lines of each transformed constructs were selected and used in the Method section. This is more important than how many T2/T3 plants were used.

The information was added in the revised section 2.7.

  1. Only one dmp8 and dmp9 mutant line was used in the study, which cannot exclude the concern of T-DNA insertion position or off target effect.

This study confirmed the deletion of T-DNA from the dmp9 line (section 2. 4). A dmp8 mutant is a T-DNA insertion line (SALK_13115; purchased from Nottingham Arabidopsis Stock Centre), and other laboratories have used the same dmp8 mutant line. Each laboratory has independently produced a dmp9 mutant line through genome editing, and a similar phenotype has been reported (Fig. 1D and S3C) (Cyprys et al. 2019; doi.org/10.1038/s41477-019-0382-3 / Wang et al. 2022), although no laboratory has checked the off-target through whole genome sequencing. Therefore, we believe that the dmp8&dmp9 mutant produced in this study is applicable for analysis.

  1. Draw schematic diagrams for each constructs as a supplemental figure for easier understanding.

All drawings are added as Figure S2.

  1. Cite reference for PIP2a and agro transformation.

Each reference was added. 

  1. Line 232-233 “Such result might reflect the heterogeneous GCS1/HAP2 expression per pollen within the pollen population”, aren’t they homozygous? Check if there are multiple T-DNAs in a line. Make sure pollen collection method is good.

We added the detailed process in the revised materials & methods. The response regarding homo/heterozygosities relates to the above Response 1.

  1. For Fig 1D, the sample number for EF and CF is too low to conclude bias fertilization or not. Also, the number of unfertilized ovules is much higher in dmp8/9 double mutants than in RNAi lines, suggesting its important and equal function in double fertilization.

Thank you for pointing that out. As the reviewer commented, dmp8&9 also causes a double fertilization block, as reflected in seed development data (Figure S3C; the undeveloped seeds reflect double fertilization or central cell fertilization failure). In Figure 1D, we wanted to indicate the biased fertilization between the female gametes was not detected in GCS1_RNAi, whereas it was observed in dmp8&9 as the control.

As shown in revised Fig. 1A, T1 transgenic lines showed increased aborted seed phenotype, implying single fertilization of the central cell occurred. The pollen produced in the T1 transgenic plant has progressed meiosis so that it was speculated that some sperm cells have altered amounts of GCS1. In addition, fertilization patterns were analyzed with ovules from T1-T3 plants. Therefore, the chance of finding the pattern of single fertilization was low. Also, the low number of single fertilization patterns could reflect that the altered GCS1 level irregularly causes it without preference for the difference of female gametes.

In this evaluation, the ovules were observed 7-9 hours after pollination (information was added in the revised version). As described in the text, two condensed sperm nuclei signals reflect the sperm cells in different states (unreleased from the pollen tube/ immediately before double fertilization/ double fertilization failure). Therefore, we shouldn’t have included the NF data as in the previous study (Takahashi et al. 2018 doi:10.1242/dev.170076). Re-calculated frequencies obtained with the ovules containing at least one fertilizing sperm cell are indicated in the revised manuscript.

  1. Line 241 and others, what do you mean by “over a fusion-required amount”?

We wanted to mean that the “amount of GCS1 barely enough for membrane fusion”. This is a speculation that the sperm with a certain amount of GCS1 on its surface can fuse with the female gamete membrane because the gcs1 sperm never progresses membrane fusion, whereas the GCS1_RNAi sperm showed single fertilization. We didn’t use the word “certain amount” because the amount was not analyzed.

  1. Line 282 and more, what do you mean by “linear mNG signal”? Did you mean plasma membrane-like localization?  For this, it’s better to do co-localization with a PM marker. For Fig S6C, the signal of TET9-mNG also looks like “perinuclear”. And, what’s the image in the right bottom corner of Fig S6 for?

We sincerely apologize that we have wrongly uploaded the previous Fig. S6 in preparation status. TET9 is a known plasma membrane marker. The photo of TET9-mNG in pollen was added in revised Fig. S7. Of course, co-localization analysis with PM marker is one of the practical approaches. Still, double labeling of different membrane proteins with green and red type FPs is a concern for in vivo sperm membrane imaging in an embryo sac. Because the cells comprising ovule tissue have plastid (chloroplast), it may be difficult to distinguish the red-type FP signal in membrane from autofluorescence form the outer tissue (in the case of histone labeling as nuclei marker, signals are condensed providing enough intensity for capturing).

  1. Line 350 and more, I think “competition” is more accurate than “antagonistic effect”: GAH (GCS1 without TM from egg cell) competes the unknown interactor with GCS1 with TM in the sperm cell before they arrive. It looks like signaling is triggered in the sperm cells (via GCS1 with TM) once GCS1-DMP8/9-UNKNOWN INTERACTOR interact with each other for egg cell fertilization.

We revised the texts according to your kind suggestion.

  1. For Fig3C, the authors show two pairs of arrested sperm nuclei, which is usually a less common phenotype compare to single pollen tube (one pair of sperm nuclei). Authors need to clarify if this is the dominant/representative phenotype or not, otherwise it could be misleading to readers.

             To avoid misleading, the photo replaced with the ovule with arrested two sperm nuclei.  

  1. Line 369 and 376, Figure S4 should be Figure S5. Line 386-387, check the Figure order or the statement, as what provided doesn’t match the statement in front.

             We are sorry for the mistakes. We corrected these in the revised manuscript.

Round 2

Reviewer 1 Report

The Authors addressed my concerns as much as they could. However considering efforts and objective difficulties on addressing full in relatively short time I would suggest pubblication as short note or similar forms   

Author Response

Dear Reviewer 1,

Thank you very much for your understanding of the publication. We further revise the manuscript, addressing the comments by reviewer 2.

Sincerely,

Tomoko Igawa

Reviewer 2 Report

The revised manuscript is much better but still needs to be improved. Here are the comments.

1.    Regarding experiments for gene expression of GCS1 in RNAi lines, the pollen collection method with acetone seems to be a concern. Gene expression can be modified upon acetone treatment (eg. https://www.ncbi.nlm.nih.gov/pmc/articles/PMC3187227/, https://ieeexplore.ieee.org/document/1137066, just list a few), it’s hard to rule out the effect of acetone on GCS1 expression. Instead, two more saver methods may be used. a). Pollen collection by vacuum (https://doi.org/10.1111/j.1365-313X.2004.02147.x, Plant Journal 2004) in T3 homozygous lines; b) entire flowers for T1 lines since it’s specifically expressed in sperm cells (https://doi.org/10.1093/jxb/erab525). Since the function of different GCS1 level in double fertilization is a very interesting biological question, editors please give the authors sufficient time to re-do the experiments.

On the other hand, even RNAi is not stable in T3, don’t remove the T3 seed set data (move to supplementary instead).

2.    For the complementation of gcs1/+ by GCS1-mNG (90%-98.2%), if not all pollen are expressing the transgene in a homozygous line (response to Point 2, and line 706), it suggests multiple insertion or something else happening. And, the number of ~90% suggests a mild defect of the truncated protein in the construct compared with the intact protein. Or, the GCS1 expression level could be a better explanation for the gradient complementation from 3 lines (as you proposed in version 1). I suggest to test it with the new pollen collection methods mentioned above.

For both A and B in FigS5, the WT control (HTR10) grown at the same time is required. For FigS5B and Table S2, Chi-square test is needed for comparing the ratio of these three phenotypes among different genotypes (or binominal test if just comparing normal vs abnormal).

3.    For single line of dmp8 or dmp9, cite the publications that show the same phenotype you observed (section 2.4).

4.    For the response to point 9, explain the same thing in the very beginning of the manuscript when firstly introducing “over a fusion-required amount”.

5.    Line 136, “refuse” -> fail to.

Author Response

Dear Reviewer 2,

Thank you very much for the comments to improve the study further. Please see our response to your comments.

We would be grateful if you could agree on the present content.

Sincerely,

Tomoko Igawa
